# The Prognostic Value of Pain Phenotyping in Relation to Treatment Outcomes in Patients with Axial Spondyloarthritis Treated in Clinical Practice: A Prospective Cohort Study

**DOI:** 10.3390/jcm10071469

**Published:** 2021-04-02

**Authors:** Rikke Asmussen Andreasen, Lars Erik Kristensen, Kenneth Egstrup, Xenofon Baraliakos, Vibeke Strand, Hans Christian Horn, Jimmi Wied, Berit Schiøttz-Christensen, Claus Aalykke, Inger Marie Jensen Hansen, Torkell Ellingsen, Robin Christensen

**Affiliations:** 1Department of Medicine, Section of Rheumatology, Odense University Hospital, Svendborg/University of Southern Denmark, 5700 Svendborg, Denmark; inger.marie.jensen.hansen@rsyd.dk; 2Section for Biostatistics and Evidence-Based Research, The Parker Institute, Bispebjerg and Frederiksberg Hospital, 2000 Copenhagen, Denmark; lars.erik.kristensen@regionh.dk (L.E.K.); robin.christensen@regionh.dk (R.C.); 3The DANBIO Registry, Centre for Rheumatology and Spine Diseases, 2100 Copenhagen, Denmark; 4Cardiovascular Research Unit, Odense University Hospital, 5700 Svendborg, Denmark; Kenneth.Egstrup@rsyd.dk; 5Rheumazentrum Ruhrgebiet Herne, Ruhr-University Bochum, 44801 Bochum, Germany; baraliakos@icloud.com; 6Division of Immunology/Rheumatology, Stanford University, Palo Alto, CA 94305, USA; vibekestrand@me.com; 7The Rheumatology Research Unit, Department of Clinical Research, Odense University Hospital, University of Southern Denmark, 5000 Odense, Denmark; hans.horn@rsyd.dk; 8Department of Ophthalmology, Odense University Hospital, 5000 Odense, Denmark; jimmi.wied@rsyd.dk; 9Institute of Regional Health Research, University of Southern Denmark, 5230 Odense, Denmark; bschiottz@health.sdu.dk; 10Department of Medicine, Section of Gastroenterology, Odense University Hospital, 5700 Svendborg, Denmark; claus.aalykke@rsyd.dk; 11OPEN, Odense Patient Data Explorative Network, Odense University Hospital, 5000 Odense, Denmark

**Keywords:** spondyloarthritis, ankylosing spondylitis, pain, prognostic factor, quality of life

## Abstract

Despite the control of inflammation, many patients with axial spondyloarthritis (axSpA) still report pain as a significant concern. Our objective was to explore the prognostic value of the painDETECT questionnaire (PDQ) in relation to treatment outcomes in axSpA patients treated in clinical practice. AxSpA patients with high disease activity initiating or switching a biological Disease-Modifying Antirheumatic Drug (bDMARD) were eligible. The PDQ score (range: −1 to 38) was used to distinguish participants with nociceptive pain (NcP) mechanisms from participants with a mixed pain mechanism (MP). The primary outcome was the proportion of individuals achieving a 50% improvement of the Bath Ankylosing Spondylitis Disease Activity Index (BASDAI50) at 12 weeks; logistic regression analysis models were used to determine the prognostic value of the nociceptive pain phenotype. Changes in continuous outcomes such as the Assessment of SpondyloArthritis International Society (ASAS) core outcome domains were analyzed using analysis of covariance (ANCOVA). Health-related quality of life (HR-QoL) was addressed using the Medical Outcomes Study SF-36. During a period of 22 months, 49 axSpA patients were included. Twenty (41%) had an NcP phenotype according to the PDQ score. BASDAI50 responses were reported by 40% (8/20) and 28% (8/29) NcP and MP groups, respectively. However, a prognostic value was not found in relation to the primary outcome (crude odds ratio [95% confidence interval]: 1.75 [0.52 to 5.87]). Across most of the secondary outcomes, axSpA NcP phenotype patients were reported having the most improvements in the HR-QoL measures. These data indicate the influence of personalized management strategies according to patients’ pain phenotypes for stratification of axSpA patients in randomized controlled trials.

## 1. Introduction

Axial spondyloarthritis (axSpA) includes both patients who have developed structural damage in the sacroiliac joints (radiographic axial spondyloarthritis (r-axSpA), also termed ankylosing spondylitis (AS)) and those without radiographic evidence of structural damage (nonradiographic axial spondyloarthritis (nr-axSpA)) [1]. Cardinal clinical signs and symptoms of axSpA include inflammatory back pain, stiffness and impaired spinal mobility [2,3]. Pain in axSpA often is assumed to be due to inflammation, not necessarily associated with inflammation nor radiographic measures of disease (e.g., elevated C-reactive protein (CRP) or bone marrow edema on sacroiliac joint MRI) [4]. Persistent pain in axSpA patients without any objective signs of inflammation is challenging for clinicians. Patients who report a large symptom burden at baseline (e.g., high Bath Ankylosing Spondylitis Disease Activity Index (BASDAI)) are more likely to have poor treatment responses [5]. There is a lack of knowledge regarding the type of pain in axSpA patients. Some studies suggest that pain other than nociceptive pain may play a role in the pain mechanism in axSpA (i.e., fibromyalgia/central pain sensitization/neuropathic pain) [6,7,8]. Recent studies state that about one-third of axSpA patients have a concomitant neuropathic pain component [8,9]. Nociceptive and neuropathic pain imply that these require different management strategies and a correct pain diagnosis informing the treatment of axSpA patients is highly desirable [10]. However, no gold standard currently exists for diagnosing neuropathic pain in a prospective clinical setting. The painDETECT questionnaire (PDQ) is a simple, validated patient-based questionnaire, developed as a screening tool to evaluate the likelihood of a present neuropathic pain (NeP) component [11].

Our primary objective was to compare the effect of initiating biological Disease-Modifying Antirheumatic Drugs (bDMARDs) on BASDAI50 responses (after 12 weeks) in axSpA patients referred with high disease activity who have a nociceptive pain (NcP) profile at baseline, compared with those with a “mixed pain” profile. Our secondary objective was to estimate reported changes in core outcome domains of axSpA from baseline to follow-up according to pain phenotypes. Thirdly, we compared treatment impact on health-related quality of life (HR-QoL) in axSpA patients in terms of how the pain phenotypes differentially affect aspects of physical, mental and social wellbeing. Finally, we explored the association between the Harvey–Bradshaw Index (HBI) [12]/Simple Clinical Colitis Activity Index (SCCAI) [13] and outcome measures reflecting disease activity at baseline and follow-up.

## 2. Materials and Methods

The study was registered before patients were enrolled (NCT02948608). As prespecified in the protocol [10] and elaborated on the Statistical Analysis Plan (SAP), it was designed as an observational cohort with the prospective enrolment of axSpA patients initiating (or switching) treatment with a biologic disease-modifying antirheumatic agent (bDMARD) due to active axSpA. The inclusion period was from 1 November 2016 to 30 September 2018.

### 2.1. Study Participants

Participants were recruited from the Department of Rheumatology, Odense University Hospital, Svendborg/Odense, and the Spine Centre of Southern Denmark, Hospital Lillebaelt, Denmark. We used the Assessment of SpondyloArthritis International Society (ASAS) classification criteria for nr-axSpA (imaging arm) [14] and the modified New York criteria for r-axSpA [15]. To be considered for inclusion, participants had to be ≥18 years, fulfill the clinical indication to initiate or switch to bDMARD treatment and the ability and willingness to give written informed consent and meet the requirements of the prespecified protocol. Participants were excluded from the study if any of the following criteria were present: current or past malignant disease, multiple sclerosis, heart failure (New York Heart Association (NYHA) class III/IV), pregnancy, diagnosed hepatitis and tuberculosis.

### 2.2. Treatment

Tumor necrosis factor α inhibitors (TNFi) were initiated in accordance with the Danish National Treatment Guidelines developed by the Danish Council for the use of expensive hospital medicines (RADS). First-line TNFI during the study period was a biosimilar Infliximab (Remsima^®^, Orion Pharma, Espoo, Finland) (27). If treatment was not tolerated, it was stopped. Second-line therapy (for those switching bDMARD) during the study period was biosimilar Etanercept (Benepali^®^, Samsung Bioepis, Incheon, Korea) administered at a dose of 50 mg subcutaneously every week. As part of clinical practice, patients were allowed to continue their conventional synthetic disease-modifying antirheumatic drugs (csDMARDs) methotrexate and/or sulfasalazine. In accordance with the Danish guidelines for intervention strategy, patients could receive nonsteroidal anti-inflammatory drugs (NSAIDs) and intra-articular glucocorticoid injections if needed during the study.

### 2.3. Data Collection

Patients completed an examination program at two time points, at baseline and after 12 weeks of treatment, according to the clinical standards in Denmark. This included an interview, questionnaires, clinical assessment, paraclinical assessments and an eye examination. Information regarding demographic, disease characteristics, medication, axSpA features, patient-reported outcome measures (PROMs) and comorbidity obtained by using the Charlson Comorbidity Index (CCI) [16] was obtained.

R.A.A. performed a clinical examination. The core domains set recommended by the ASAS/Outcome Measures in Rheumatology (OMERACT) including physical function, pain, spinal mobility, spinal stiffness, patient’s global assessment (PGA), peripheral joints/entheses, acute-phase reactants and fatigue were assessed at each study visit using the instruments explained below. At the follow-up visit, PROMs, a clinical examination and laboratory tests were obtained. Reasons for withdrawal of bDMARD were registered.

### 2.4. PainDETECT Questionnaire

The painDETECT questionnaire (PDQ) is a self-administered questionnaire that is completed at baseline to determine whether a neuropathic pain component (NeP) is present. The PDQ is composed of questions regarding pain intensity. It consists of three numeric rating scales (NRSs): a pain course pattern, a pain drawing reflecting pain radiation and seven questions addressing somatosensory phenomena [11]. The score ranges between −1 and 38. A PDQ score <13 indicates pure nociceptive pain (NcP), a score of 13–18 is considered to reflect a mixed pain (MP) mechanism and a score ≥19 indicates that NeP is likely [8,11]. We decided to apply the PDQ score as a binary categorical variable, PDQ < 13 and PDQ ≥ 13, to distinguish participants with pure NcP mechanisms from participants with MP mechanisms. This categorization of the PDQ has been previously been applied in a study exploring pain mechanisms in psoriatic arthritis patients [17].

### 2.5. Patient-Reported Outcome Measures (PROMs) and Clinical Examination

The Bath Ankylosing Spondylitis Disease Activity Index (BASDAI) was used for the evaluation of patient disease activity [18]. Physical function (PF) was addressed using the Bath Ankylosing Spondylitis Functional Index (BASFI) [19]. Pain was assessed by a Visual Analogue Scale (VAS) referring to pain in the previous week due to axSpA on a 0–10 cm scale. Spinal stiffness (SS) was measured on a VAS scale for morning stiffness (0–120 min) [20]. The patient’s global assessment (PGA) was assessed by a single question with a range from 0 to 10 cm (VAS) [20], and fatigue was assessed by a VAS scale (0–10 cm). Health-related quality of life (HR-QoL) was addressed using the Medical Outcomes Study SF-36 to compare various aspects of health status across a general and broad patient population [21,22,23]. The physical (PCS) and mental component summary scores (MCS) were calculated. We used the Danish version of SF-36 version 2 [24]. Furthermore, we used “spydergrams” to provide a visual method to examine the eight domains of SF-36 scored from 0 (worst) to 100 (best) simultaneously in a single figure [25]. Danish normative data were used for comparisons in this specific cohort (32). Using spydergrams enabled us to determine how pain profiles differentially affect HR-QoL.

Spinal mobility (SM) was addressed using the Bath Ankylosing Spondylitis Metrology Index. [26]. Peripheral joints and entheses were addressed by registering the number of swollen joints (44-joint count) and by using a validated enthesitis score [27] (the Spondyloarthritis Research Consortium of Canada (SPARCC) enthesitis index). Acute-phase reactants (APRs) were addressed by using C-reactive protein (CRP). Furthermore, plasma and fecal calprotectin levels were measured using an ELISA kit.

### 2.6. Inflammatory Bowel Disease (IBD) Related Questionnaires

Two questionnaires regarding symptoms of inflammatory bowel disease (IBD) were used at baseline and follow-up: the Harvey–Bradshaw Activity Index (HBI) [12] and the Simple Clinical Colitis Activity Index (SCCAI) [13]. The HBI is a simpler version of the Crohn’s disease activity index (CDAI) to facilitate its usage in a daily clinical setting. The SCCAI was developed to aid the initial evaluation of exacerbations of colitis [13].

### 2.7. Eye Examination

The eye examination was performed 2–4 weeks after the baseline visit by an ophthalmologist in accordance with normal clinical practice in Denmark. The following symptoms of acute anterior uveitis were registered: pain, redness of the globe, photophobia and reduced visual acuity. Best-corrected visual acuity determined by Snellen, intraocular pressure, slit-lamp examination and dilated fundus examination were performed, and previous signs of anterior uveitis (for example, keratic precipitates and posterior synechiae) were recorded. Ocular inflammation (if present) was graded according to the Standardization of Uveitis Nomenclature working group recommendations [28].

### 2.8. Outcome Responder Criteria

Responses to treatment were primarily evaluated according to BASDAI responses. Secondary outcomes include achievement of clinically important improvements (defined as an improvement in the Ankylosing Spondylitis Disease Activity Score (ASDAS-CRP) of ≥1.1) and a major improvement (defined as ΔASDAS-CRP ≥ 2.0), as well as a change in core outcome domains.

### 2.9. Involvement of Patient Research Partners

This project followed the EULAR recommendations for patient research partners (PRPs) in scientific research [29]. The study was designed with assistance from two Danish PRPs (BD and LKF).

### 2.10. Statistical Methods

As described in the original protocol [10] and the statistical analysis plan (SAP), the anticipated analyses were outlined before reviewing the actual data. Sample size considerations.

For a comparison of two independent binomial proportions (high vs. other PDQ category), Pearson’s χ^2^ statistic was used with χ^2^ approximation, with a two-sided significance level of 0.05 and a total sample size of 54 patients with axSpA, assuming that the proportion of patients with a high PDQ of 50% achieves a statistical power of at least 80% when the proportions having a BASDAI response are 15% and 50% higher versus the other PDQ category. Thus, we aimed to include 60 patients in total (anticipating 30 patients will have a PDQ > 13), corresponding to an approximate power of 84.3% when the proportions with a BASDAI response are 15% and 50%, respectively.

The characteristics of the participants were described for each pain profile: Nociceptive Pain (NcP) and “Mixed Pain” (MP) phenotypes defined by PDQ scores <13 and ≥13, respectively. For descriptive statistics, medians and interquartile ranges (IQRs) for continuous data and absolute numbers with corresponding percentages for binary outcomes are presented. For continuous outcomes, changes were adjusted for baseline value and compared using analysis of covariance (ANCOVA). The primary analyses were based on the Intention-to-Treat (ITT) population. In accordance with the ITT principle, all participants who signed the informed consent and with a PDQ evaluation at baseline were included in the analyses regardless of their subsequent adherence to the study protocol. Missing data were handled with a single-imputation (nonresponder) technique. All *p*-values and 95% confidence intervals are two-sided. We did not apply explicit adjustments for multiplicity; instead, the results are interpreted with caution as exploratory findings in the context of multiplicity. The prognostic value of PDQ classification was examined by logistic regression models using STATA version 16 and presented as odds ratios (ORs) with the corresponding 95% confidence interval (CI). According to the SAP, the model was adjusted for age (in years), sex (male or female) and axSpA disease classification (r-axSpA or nr-axSpA).

On an exploratory basis, we determined the association between HBI/SCCAI and outcome measures reflecting disease activity (i.e., BASDAI/ASDAS-CRP) by using Spearman’s rank correlation coefficients; for consistency in presentation, these were stratified on pain profiles as well.

## 3. Results

As illustrated in Figure 1, out of 55 screened patients, 49 patients diagnosed with axSpA were included. Twenty (41%) fulfilled the criterion for an NcP phenotype. All participants completed the PDQ at baseline; 44 initiating bDMARD and 5 switching bDMARD.

### 3.1. Baseline Characteristics

Baseline characteristics are described according to pain profile in Table 1.

Participants with a MP profile presented with worse BASDAI, BASFI, pain, VAS-global, SF-36 PCS, more tender joints and tender points compared at baseline, with those with pure NcP profiles who presented with higher calprotectin in plasma compared with participants with MP. No differences in demographics, medication, comorbidities and prevalence of HLA-B27 were observed. We did not find a statistically significant difference across the two PDQ classification groups in the prevalence of fibromyalgia. Only two patients were found to have signs of previous uveitis at the eye examination (posterior synechiae), and both were HLA-B27 positive. None of the examined showed signs or symptoms of active inflammation.

### 3.2. BASDAI Responses and Changes in Core Outcome Domains

Table 2 shows the BASDAI responses and changes in core outcome domains from baseline-follow-up. BASDAI50 responses were reported by 40% (8/20) axSpA patients with NcP profile compared with 28% (8/29) MP. Nociceptive pain phenotypes did not prognosticate BASDAI50 responses, with an odds ratio (OR) of 1.75 (95% CI: 0.52 to 5.87). Although patients with a NcP phenotype apparently have a more favorable clinical outcome, no statistically significant differences across the pain profile groups were found in the proportion achieving clinically important improvement or major improvement, with crude odds ratios of 2.13 (95% CI: 0.67 to 6.78) and 1.13 (95% CI: 0.32 to 3.95), respectively.

Changes in core outcome domains from baseline to follow-up did not differ significantly between pain profiles, with the exception of spinal mobility (SM), where the MP profile group reported better SM improvements compared with NcP (−1.3 (−1.6 to −0.9), *p* = 0.035).

### 3.3. Health-Related Quality of Life

Mean SF-36 domain scores for the enrolled axSpA patients are illustrated in Figure 2. Large reductions were reported in all domains of the SF-36 at baseline compared with age- and gender-matched Danish normative values. The differences were most evident in axSpA patients with a MP profile resulting in large decrements across all eight domains. By visual inspection, the changes in the domain GH appeared less evident in the NcP group compared with the MP group. However, both groups reported am improvement, meeting minimum clinically important differences, defined as ≥5 points in individual SF-36 domain scores. After 12 weeks of treatment with a bDMARD, all eight domains improved in both pain profiles. However, improvements were greater in patients with pure NcP profiles. Most improvements were reported in domains role emotional (RE), SF and mental health (MH) in the NcP group, increasing to approach Danish norms.

### 3.4. Associations between Baseline HBI and SCCAI and Variables Reflecting Disease Activity

The baseline HBI scores (ranging from 0 to 16+) were not associated with BASDAI, ASDAS-CRP, CRP or calprotectin for any of the pain profiles (Appendix A). Increasing baseline SCCAI showed a moderate negative statistical correlation with fecal calprotectin for the NcP profile group (Appendix A).

## 4. Discussion

In our study, forty-one percent of axSpA patients presented with pure NcP. This was lower than previous studies of pain mechanisms in axSpA [9,30]. A Danish nationwide study reported that 55% of patients with spondyloarthritis have pure NcP (PDQ < 13). However, the proportion of axSpA patients in this group is unclear [30]. Another study reported that 75% of their axSpA patients presented with pure NcP. However, they only included r-axSpA patients with stable disease, and furthermore, they excluded participants if they had fibromyalgia or depression [9]. To our knowledge, this present study is the first prospective study to evaluate the prognostic value of pain profiles by the PDQ score in relation to treatment responses in axSpA patients initiating or switching bDMARDs.

We anticipated that the response rate to bDMARD therapies would be different according to pain profiles. However, this was surprisingly not confirmed. At baseline, we found statistically significant differences across the two PDQ classification groups in BASDAI, BASFI, VAS-pain, VAS-global, tender joint count (TJC), tender point count (TPC) and SF-36-PCS. These variables have been shown to be associated with (or being a proxy for) widespread pain/NeP/fibromyalgia in axSpA [31]; therefore, it was surprising that we did not find any differences in the proportion reporting treatment responses according to pain profiles. We know that smoking has an impact on treatment response. The DANBIO Danish nationwide registry found that smokers had odds of approximately 0.5 in meeting BASDAI50 response criteria in comparison with nonsmokers [32]. The proportion of current smokers was high in the NcP group and could possibly have confounded treatment responses. However, the responses were comparable to other studies investigating TNFI treatment in axSpA patients with high disease activity [33].

The PDQ did not have a prognostic value in relation to BASDAI50 responses or change in ASDAS-CRP in the regression analyses. However, in light of the small sample size, it does not seem reasonable to reject any prognostic value of the PDQ. A positive PDQ score still attracts the clinician’s attention to the possibility of neuropathic pain component and encourages performing an adequate neurological examination and the consideration of further testing when needed. Although we could not do as such, a proportion of axSpA patients would possibly benefit from the management of non-nociceptive pain with targeted treatment. Further large-scale studies are needed to clarify the prognostic value of PDQ in axSpA patients.

Our study confirms that HR-QoL is much reduced in patients with axSpA compared with Danish norms. An interesting observation from our data was that both baseline and follow-up patterns of HR-QoL appear different in each pain profile group, as illustrated by the spydergrams. The patients with a NcP profile reported the greatest improvement in all the domains compared with those with a MP component.

The prevalence of sequelae of uveitis in our axSpA population was low. It may be the reflection of increasing awareness of the importance of early diagnosis and subsequent treatment of uveitis among axSpA patients. We did not find any association between baseline questionnaires regarding symptoms of IBD and measures reflecting axSpA disease activity. It was surprising that SCCAI was negatively correlated with fecal calprotectin levels in axSpA patients classified with pure NcP. We found differences between NcP and MP in p-calprotectin. Calprotectin is mainly expressed on leukocytes, in circulating neutrophils and monocytes and in the neutrophils and macrophages of inflamed tissue. In the presence of an ongoing cycle of inflammation, calprotectin levels are increased in plasma and synovial fluid. The level of p-calprotectin differs significantly between the two pain groups, which may reflect the burden of inflammation somehow might be higher in the NcP group.

The main strength of this study was the prospective design with a prespecified protocol and statistical analysis plan. Thereby, outcome reporting bias was reduced. Another strength was the use of the ASAS/OMEARCT-endorsed outcome domains. A major concern using cohort study designs is the risk of confounding bias; unlike RCTs, where there is an unbiased distribution of confounding. A limitation of our study was the small study sample size, which restricted the possibility of investigating other potential confounders such as smoking, BMI, bionaive status and educational levels.


## 5. Conclusions

Even though a mixed pain profile was present in nearly two-thirds of patients with axSpA in this consecutively sampled cohort, associated with worse BASDAI, BASFI, pain, VAS-global, SF-36 PCS levels, and more tender joints at enrolment, it did not have an impact on treatment responses. This could potentially be due to a type-2 error because the signal was there, with potential for personalized medicine, although without statistical power to detect the difference between groups likely because of the sample size. In other words, the PDQ score did not have a prognostic value in relation to achieving BASDAI50 responses or changes in ASDAS-CRP per se. However, further large-scale studies could potentially provide evidence for use of the pain phenotype as a screening model in clinical practice to identify axSpA patients with extra needs.

## Figures and Tables

**Figure 1 jcm-10-01469-f001:**
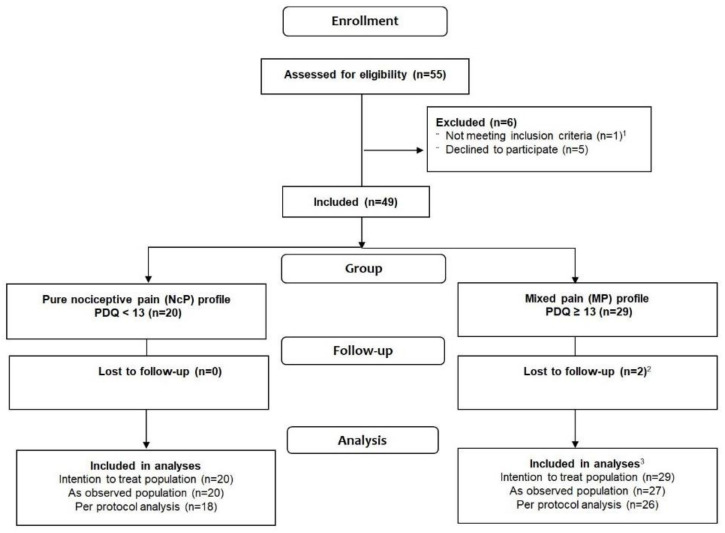
Flow chart. ^1^ Did not fulfill axSpA classification criteria. ^2^ Reasons for drop out: one participant moved to another hospital, and one participant was hospitalized. ^3^ As observed refers to participants with available data for analyses and per protocol to participants with available data for analyses and adherence to the prespecified protocol.

**Figure 2 jcm-10-01469-f002:**
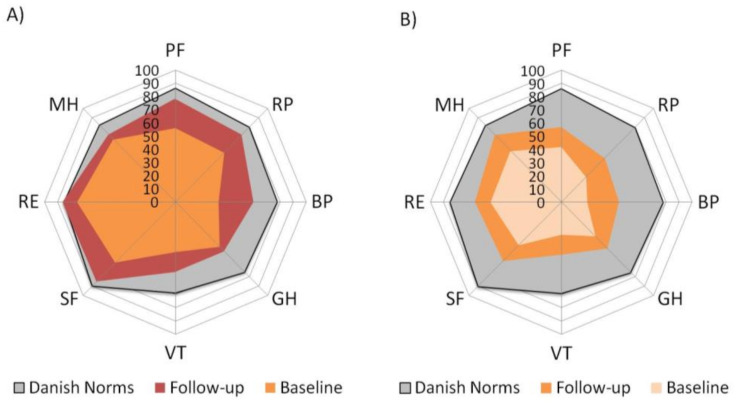
Mean SF-36 scores for Danish axSpA patients stratified by pain profiles. (**A**) axSpA patients classified as pure nociceptive pain (NcP) profile at baseline (**B**) axSpA patients classified as mixed pain (MP) profile. Mean SF-36 scores for Danish norms are also shown. PF, physical function; RP, role physical; BP, bodily pain; GH, general health; VT, vitality; SF, social functioning; RE, role emotional; MH, mental health.

**Table 1 jcm-10-01469-t001:** Baseline characteristics according to pain phenotype, nociceptive pain (NcP) and mixed pain (MP).

Demographics	NcP (*n* = 20)	MP (*n* = 29)	*p*-Value
Males, no. (%)	12 (60)	13 (45)	0.387 ^a^
Age, years, median (Q_1_; Q_3_)	49 (40; 58)	43 (32; 51)	0.106 ^b^
PDQ score, median (Q_1_; Q_3_)	7 (4.5; 9)	19 (16; 21)	<0.001 ^b^
BMI, kg/m^2^, median (Q_1_; Q_3_)	26.1 (24.4; 28.6)	26.6 (24.2; 30.5)	0.483 ^b^
Smoking (current), no. (%)	9 (45)	9 (31)	0.375 ^a^
AxSpA symptom duration, months, median (Q_1_; Q_3_)	109 (22; 229)	50 (27; 84)	0.489 ^b^
Radiographic axSpA, no. (%)	10 (50)	6 (21)	0.061 ^a^
Higher education, no. (%)	13 (65)	15 (52)	0.394 ^a^
Peripheral joint involvement, no. (%)	11 (55)	13 (45)	0.567 ^a^
**Comorbidity**			
CCI, median (Q_1_; Q_3_)	1 (0; 2)	1 (0; 1)	0.705 ^b^
Mental disorder (depression, anxiety) past/present, no. (%)	2 (10)	10 (34)	0.089 ^a^
**Medication**			
NSAIDs daily use, no. (%)	18 (90)	24 (83)	0.685 ^a^
MTX use, no. (%)	7 (35)	9 (31)	1.000 ^a^
MTX dose (mg/week), median (Q_1_; Q_3_)	0 (0; 7.5)	0 (0; 15.0)	0.942 ^b^
SSZ use, no. (%)	2 (10)	3 (10)	1.000 ^a^
bDMARD I, no. (%)	19 (95)	25 (76)	0.636 ^a^
Glucocorticoid use, no. (%)	1 (5)	0 (0)	0.408 ^a^
Glucocorticoid dose (mg/day) median (Q_1_; Q_3_)	0 (0; 0)	0 (0; 0)	0.229 ^b^
**Extra-articular manifestations ***			
Uveitis, no. (%)	5 (25)	4 (14)	0.456 ^a^
IBD, no. (%)	0 (0)	2 (7)	0.507 ^a^
Psoriasis, no. (%)	4 (20)	10 (34)	0.345 ^a^
Dactylitis, no. (%)	7 (35)	12 (41)	0.769 ^a^
Achilles-enthesitis, no. (%)	13 (65)	23 (79)	0.331 ^a^
Nephrolithiasis, no. (%)	4 (20)	6 (21)	1.000 ^a^
**Patient-reported outcome measures (PROMs)**			
BASDAI (0–10), median (Q_1_; Q_3_)	57 (46; 70)	77 (64; 85)	0.002 ^b^
BASFI (0–10), median (Q_1_; Q_3_)	44 (33; 53)	67 (58; 77)	<0.001 ^b^
VAS pain (0–10), median (Q_1_; Q_3_)	65 (53; 79)	77 (69; 84)	0.026 ^b^
VAS fatigue (0–10), median (Q_1_; Q_3_)	72 (55; 83)	81 (74; 86)	0.060 ^b^
VAS global (0–10), median (Q_1_; Q_3_)	73 (61; 87)	83 (74; 92)	0.033 ^b^
SF36-MCS (0–100), median (Q_1_; Q_3_)	47.6 (37.8; 52.0)	41.3 (36.2; 49.9)	0.382 ^b^
SF36-PCS (0–100), median (Q_1_; Q_3_)	38.0 (35.0; 44.8)	30.3 (28.5; 36.5)	0.002 ^b^
HBI (0–16+), median (Q_1_; Q_3_)	5 (4; 6)	4 (4; 6)	0.491 ^b^
SCCAI (0–19), median (Q_1_; Q_3_)	4 (3; 5)	4 (3; 4)	0.892 ^b^
**Clinical examination**	NcP (*n* = 20)	MP (*n* = 29)	*p*-value
BASMI (0–10), median (Q_1_; Q_3_)	20 (15; 35)	20 (10; 30)	0.892 ^b^
Tender joint count; TJC (0–44), median (Q_1_; Q_3_)	2 (1; 4)	6 (2; 8)	0.020 ^b^
Swollen joint count; SJC (0–44), median (Q_1_; Q_3_)	0 (0; 0)	0 (0; 0)	0.965 ^b^
Fibromyalgia, no. (%)	1 (5)	5 (17)	0.199 ^a^
Tender point count; TPC (0–18), median (Q_1_; Q_3_)	2 (0; 3)	5 (2; 9)	0.023 ^b^
SPARCC enthesitis index (0–16), median (Q_1_; Q_3_)	2 (0; 3)	2 (1; 6)	0.098 ^b^
ASDAS-CRP, median (Q_1_; Q_3_)	2.2 (1.3; 2.9)	2.9 (1,8; 3.6)	0.063 ^b^
VAS physician (0–rwe10), median (Q_1_; Q_3_)	54 (44; 63)	62 (44; 72)	0.213 ^b^
**Paraclinical assessment**			
HLA-B27, no. (%)	15 (75)	17 (59)	0.213 ^a^
CRP (mg/L), median (Q_1_; Q_3_)	8.7 (2.5; 14.0)	6 (1; 12)	0.450 ^b^
P-calprotectin (μg/mL), median (Q_1_; Q_3_)	325 (234.9; 365)	195 (153; 324)	0.035 ^b^
F-calprotectin (<50 × 10^−6^), median (Q_1_; Q_3_)	34 (13; 77)	16 (14; 41)	0.212 ^b^

PDQ, painDETECT questionnaire; BMI, body mass index; CCI, Charlson Comorbidity Index; NSAIDs, non-steroidal anti-inflammatory drugs; MTX, methotrexate; SSZ, sulfasalazine; bDMARD, biologic disease-modifying drugs; IBD, inflammatory bowel disease; BASDAI, Bath Ankylosing Spondylitis Disease Activity Index; BASFI, Bath Ankylosing Spondylitis Functional Index; VAS, Visual Analogue Scale; SF-36: MCS/PCS, Medical Outcomes Study Short Form 36 Mental/Physical Component Summary; HBAI, Harvey–Bradshaw Activity Index; SCCAI, Simple Colitis Activity Index; BASMI, Bath Ankylosing Spondylitis Metrology Index; SPARCC, Spondyloarthritis Research Consortium of Canada; ASDAS-CRP, Ankylosing Spondylitis Disease Activity Score—C-reactive Protein; P/F-Calprotectin, plasma/fecal calprotectin, * either patient history or current diagnosis. ^a^ Fisher’s exact test (two-sided). ^b^ Kruskal–Wallis *H* test. * Either patient history or current diagnosis.

**Table 2 jcm-10-01469-t002:** Change in BASDAI responses and axSpA core domains stratified by pain phenotype at enrolment in the Intention-to-Treat (ITT) population.

Outcome	NcP (*n* = 20)	MP (*n* = 29)	OR, Crude (95% CI)	OR, Adjusted * (95% CI)	*p*-Values Crude/Adjusted *
BASDAI50 responders, no. (%)	8 (40)	8 (28)	1.75 (0.52 to 5.87)	1.23 (0.33 to 4.6)	0.362/0.221
ΔBASDAI ≥ 2, no. (%)	11 (55)	12 (41)	1.73 (0.55 to 5.47)	0.94 (0.24 to 3.67)	0.348/0.09
ΔASDAS-CRP ≥ 1.1, no. (%)	12 (60)	12 (41)	2.13 (0.67 to 6.78)	1.22 (0.36 to 4.20)	0.200/0.362
ΔASDAS-CRP ≥ 2.0, no (%)	6 (30)	8 (28)	1.13 (0.32 to 3.95)	0.81 (0.20 to 3.17)	0.854/0.203
**Outcome**	**NcP (*n* = 20)**	**MP (*n* = 29)**	**Contrast between Groups (95% CI)**		***p*-Value**
ΔBASDAI, score	−2.7 (−3.9 to −1.5)	−2.2 (−3.1 to −1.2)	0.5 (−1.07 to 2.18)		0.496
%BASDAI change, %	−40 (−56 to −23)	−32 (−46 to −18)	8.0 (−16 to 30)		0.522
ΔASDAS-CRP, score	−1.2 (−1.6 to −0.9)	−1.6 (−1.9 to 1.3)	−0.40 (−0.83 to 0.1)		0.088
**Health-related quality of life**					
ΔSF36-PCS (0–100)	8.5 (3.8 to 13.2)	4.4 (0.4 to 8.4)	−4.1 (−10.7 to 2.37)		0.206
ΔSF36-MCS (0–100)	8.1 (3.9 to 12.3)	4.4 (0.8 to 8.0)	−3.70 (−9.2 to 1.8)		0.185
**Core domains**					
ΔPF (BASFI 0–10)	−2.1 (−3.2 to −0.9)	−1.6 (−2.5 to −0.6)	0.50 (−1.1 to 2.1)		0.532
ΔPain (VAS 0–10)	−3.3 (−4.6 to −2.0)	−2.3 (−3.4 to −1.2)	1.0 (−0.8 to 2.8)		0.255
ΔSM (BASMI 0–10)	−0.7 (−1.0 to −0.3)	−1.3 (−1.6 to −0.9)	−0.6 (−1.1 to −0.4)		0.035
ΔSS (VAS 0–10)	−4.0 (−5.3 to −2.7)	−2.9 (−4.0 to 1.8)	1.1 (−0.7 to 2.8)		0.215
ΔPGA (VAS 0–10)	−2.9 (−4.1 to −1.6)	−2.6 (−3.6 to −1.6)	0.3 (−1.4 to 1.9)		0.754
ΔSwollen joint count (0–44)	−0.5 (−0.7 to −0.4)	−0.6 (−0.7 to 0.5)	−0.1 (−0.2 to 0.1)		0.400
ΔEnthesitis Index (0–18)	−1.5 (−2.3 to −0.8)	−1.7 (−2.3 to −1.0)	−0.2 (−1.2 to 0.9)		0.734
ΔAPR (CRP, mg/L)	−9.6 (−24.1 to 5.1)	−2.0 (−14.1 to 10.1)	7.6 (−11.4 to 26.5)		0.429
ΔFatigue (VAS 0–10)	−1.9 (−3.2 to −0.6)	−1.8 (−3.0 to −7.2)	−0.1 (−1.7 to 1.8)		0.937

Group statistics are reported as least-squares means (95% CI) unless otherwise stated. BASDAI50 response: 50% reduction in BASDAI, Bath Ankylosing Spondylitis Disease Activity Index, to ASDAS-CRP, Ankylosing Spondylitis Disease Activity Score, to CRP, C-Reactive Protein, to SF36-PCS/MCS, Medical Outcomes Study Short Form 36 Physical/Mental Component Summary, to PF, physical function, to SM, spinal mobility, to SS, spinal stiffness, to PGA, patient’s global assessment, to APR, to acute-phase reactant. Dichotomous outcomes were compared using logistic regression, reported as the odds ratio. * adjusted for sex, age and axSpA disease classification. Continuous outcomes were compared using analysis of covariance adjusting for the level at baseline. Analyses are reported as the difference between least-squares means.

## Data Availability

Available on reasonable request.

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
