# Peer review of "The Prognostic Value of Pain Phenotyping in Relation to Treatment Outcomes in Patients with Axial Spondyloarthritis Treated in Clinical Practice: A Prospective Cohort Study"

_jcm, 2021, doi:10.3390/jcm10071469_

Round 1

Reviewer 1 Report

This is an interesting paper for rheumatologists. I have some comments.

1. There are no significant differences, but there are clear differences between NcP and MP in axSpA symptom duration, radiographic axSpA number, and mental disorder number. Do the authors have any thoughts on how these affect the results?

2. Please discuss why there are significant differences between NcP and MP in p-calprotectin.

3. What is the reason for the poor GH improvement in NcP in Figure 2?

Author Response

Author response: JCM-1128206                                                      
The prognostic value of pain phenotyping in relation to treatment outcomes in patients with axial spondyloarthritis treated in clinical practice: A prospective cohort study

Dear Editor,

Please receive this revised version of our manuscript.

We appreciate the insightful comments from the reviewers and the editorial board. Below you will find a point-by-point account of how we addressed the different issues.

Changes are highlighted with the ‘track changes function’ in Word. We are very grateful to the suggestions made, and we think that this revised version is an improvement of the previous version. We hope you find this version acceptable for publication. If you have further queries or questions, please do not hesitate to contact us.

Best regards,

Rikke A. Andreasen, Torkell Ellingsen & Robin Christensen on behalf of the entire author group.

Response to Reviewer 1 Comments

Point 1: There are no significant differences, but there are clear differences between NcP and MP in axSpA symptom duration, radiographic axSpA number, and mental disorder number. Do the authors have any thoughts on how these affect the results?

Response 1: Thank you for this comment. We did not find a statistically difference between NcP and MP in axSpA symptom duration, radiographic axSpA number, and mental disorder number, and it did not affect the results in our study. One could only speculate if the axSpA patients classified as r-axSpA (and thereby also longer symptom duration) are more likely to have pure nociceptive pain and less mental disorders compared with the MP group. However, patients with chronic pain (regardless of the underlying cause of pain) have been reported to have a greater prevalence of mental disorders and somatization than that found in the general population, but the true association between pain/type of pain and mental disorders is still unknown.

Point 2: Please discuss why there are significant differences between NcP and MP in p-calprotectin.

Response 2:

Sorry, we missed to explain this observation. We have now added the following to the discussion section (p. 11).

‘Calprotectin is mainly expressed on leukocytes, in circulating neutrophils and monocytes and in neutrophils and macrophages of inflamed tissue. In the presence of an ongoing cycle of inflammation, calprotectin levels are increased in plasma and synovial fluid. The level of p-calprotectin differs significantly between the two pain groups which may reflect the burden of inflammation somehow might be higher in the NcP group’.  

Point 3: What is the reason for the poor GH improvement in NcP in Figure 2?

Response 3: Thank you for your comment. The Medical Outcomes Study Short Form-36 (SF-36) was developed to measure self-reported health-related quality of life (HR-QoL). 36 questions combined into eight domains reflecting different dimensions of health.

In contrast to displaying SF-36 as eight-columned bar charts, spydergrams offer the ability to view changes more easily across all domains as a pattern recognition profile, depicting disease and population-specific “patterns” of decrements in baseline values compared with matched normative data, as well as treatment-associated changes. In our study, all eight domains improved in both pain profiles after 12 weeks of treatment with a biological agent. Most improvements were reported in domains role emotional (RE), social functioning (SF) and mental health (MH) in the NcP group. By visual inspection, the changes in the domain general health (GH) appeared less evident in the NcP group compared with the MP group. However, both pain groups reported improvement meeting minimum clinically important differences defined as ≥ 5 points in individual SF-36 domain scores.

We have elaborated further on the spydergrams in the text (p. 9).

Reviewer 2 Report

This is an interesting study examining the hypothesis than nociceptive as compared to mixed back pain might be linked to differential response to bDMARDs in patients with AxSpA. My comments are listed below.

  1. Please double check acronyms as they need to be spelled out upon first occurrence in the text.
  2. I miss some power analysis / sample size estimation based on the primary hypothesis of the study. This is critical as the sample could be insufficient to identify between-groups differences.
  3. The two main groups (NcP/MP) seem to differ in the rates of biologically naive disease. Can the authors comment? Would it be reasonable to perform a separate analysis only in bDMARD-naive patients?
  4. Can the authors comment on the performance of painDETECT and whether results should be confirmed with the use of a 2nd pain assessment test.
  5. Was fibromyalgia considered as a comorbidity in the patient cohort? Was it an exclusion criterion?

Author Response

Author response: JCM-1128206                                                      
The prognostic value of pain phenotyping in relation to treatment outcomes in patients with axial spondyloarthritis treated in clinical practice: A prospective cohort study

Dear Editor,

Please receive this revised version of our manuscript.

We appreciate the insightful comments from the reviewers and the editorial board. Below you will find a point-by-point account of how we addressed the different issues.

Changes are highlighted with the ‘track changes function’ in Word. We are very grateful to the suggestions made, and we think that this revised version is an improvement of the previous version. We hope you find this version acceptable for publication. If you have further queries or questions, please do not hesitate to contact us.

Best regards,

Rikke A. Andreasen, Torkell Ellingsen & Robin Christensen on behalf of the entire author group.

Response to Reviewer 2 Comments

Point 1:  Please double check acronyms as they need to be spelled out upon first occurrence in the text.

Response 1: That is an important notion, thank you. We have now checked our acronyms throughout our manuscript, and made sure to spell them out upon first occurrence in the text (see ‘track changes’ in Word)

Point 2: I miss some power analysis / sample size estimation based on the primary hypothesis of the study. This is critical as the sample could be insufficient to identify between-groups differences.

Response 2: Thank you for this important consideration. We performed a statistical analysis plan prior to our analysis (also uploaded as Appendix A) including sample size estimation. Despite of much effort our estimated sample size was unfortunately not met due to our scheduled timeline (PhD time line). We have now added a section ‘sample size considerations’ in the manuscript.

We have now added sample size considerations in the text (p. 4):

Sample size considerations

For a comparison of two independent binomial proportions (high vs other PDQ category) using Pearson’s χ2 statistic with a χ2 approximation with a two-sided significance level of 0.05, a total sample size of 54 patients with AxSpA assuming that the proportion of patients with a high PDQ of 50% achieves a statistical power of at least 80% when the proportions having a BASDAI response are 15% and 50%, in high versus other PDQ category. Thus, we aim to include 60 patients in total (anticipating 30 patients will have a PDQ>13), corresponding to an approximate power of 84.3% when the proportions with a BASDAI-response are 15% and 50%, respectively.

Point 3: The two main groups (NcP/MP) seem to differ in the rates of biologically naive disease. Can the authors comment? Would it be reasonable to perform a separate analysis only in bDMARD-naive patients?

Response 3: Thank you for your comment. A major concern using cohort study designs is the risk of confounding bias; unlike RCTs where there is an unbiased distribution of confounding. The number of biologically naive patients in our study seems to differ between the two groups (NcP/MP) and could potentially cofound the results. However, only five patients (one in the NcP group and four in the MP group) were biologically naïve. The small sample size restricted our ability to investigate potential confounding, such as smoking, obesity, education level, mental disorder and bDMARD naive status, which limits the overall interpretation of findings. We have elaborated on the study limitations in the text (p. 11).  

Point 4: Can the authors comment on the performance of painDETECT and whether results should be confirmed with the use of a 2nd pain assessment test.

Response 4: Thank you for your comment. We have now elaborated on that in the discussion section on page 11:

‘The PDQ did not have a prognostic value in relation to BASDAI50 responses or change in the regression analyses. However, in the light of the small sample size, it does not seem reasonable to reject any prognostic value of the PDQ. A positive PDQ score still attracts the clinician’s attention to the possibility of neuropathic pain component and encourages performing an adequate neurological examination and the consideration of further testing when needed. Although we could not such, a proportion of axSpA patients would possibly benefit from management of non-nociceptive pain with targeted treatment. Further large-scale studies are needed to clarify the prognostic value of PDQ in axSpA patients’.

Point 5: Was fibromyalgia considered as a comorbidity in the patient cohort? Was it an exclusion criterion?

Response 5:

Thank you for this important comment. We how now added the proportion of axSpA patients fulfilling the fibromyalgia criteria in Table 1 (p. 7), and added the results in the results section (p. 8). Concomitant fibromyalgia was not an exclusion criterion for our study. We did not find a statistically difference between the NcP and MP group.

Round 2

Reviewer 2 Report

The authors have addressed my points and I am content with the manuscript.